# Use of drugs for hyperlipidaemia and diabetes and risk of primary and secondary brain tumours: nested case–control studies using the UK Clinical Practice Research Datalink (CPRD)

Jamie W Robinson,[1] Richard Martin,[1,2,3] Mio Ozawa,[4]
Martha Maria Christine Elwenspoek [2,5] Maria Theresa Redaniel [2,5]
Kathreena M Kurian,[1,6] Yoav Ben-Shlomo [2,5]

KMK and YB-S are joint senior authors.

For numbered affiliations see end of article.

**Correspondence to**
Professor Yoav Ben-Shlomo;
Y.Ben-Shlomo@bristol.ac.uk

## ABSTRACT

**Objectives** Previous studies have suggested that fibrates and glitazones may have a role in brain tumour prevention. We examined if there is support for these observations using primary care records from the UK Clinical Practice Research Datalink (CPRD).

**Design** We conducted two nested case–control studies using primary and secondary brain tumours identified within CPRD between 2000 and 2016. We selected cases and controls among the population of individuals who had been treated with any anti-diabetic or anti-hyperlipidaemic medication to reduce confounding by indication.

**Setting** Adults older than 18 years registered with a general practitioner in the UK contributing data to CPRD.

**Results** We identified 7496 individuals with any brain tumour (4471 primary; 3025 secondary) in total. After restricting cases and controls to those prescribed any anti-diabetic or anti-hyperlipidaemic medication, there were 1950 cases and 7791 controls in the fibrate and 480 cases with 1920 controls in the glitazone analyses. Longer use of glitazones compared with all other anti-diabetic medications was associated with a reduced risk of primary (adjusted OR (aOR) 0.89 per year, 95% CI 0.80 to 0.98), secondary (aOR 0.87 per year, 95% CI 0.77 to 0.99) or combined brain tumours (aOR 0.88 per year, 95% CI 0.81 to 0.95). There was little evidence that fibrate exposure was associated with risk of either primary or secondary brain tumours.

**Conclusions** Longer exposure to glitazones was associated with reduced primary and secondary brain tumour risk. Further basic science and population-based research should explore this finding in greater detail, in terms of replication and mechanistic studies.

## STRENGTHS AND LIMITATIONS OF THIS STUDY

⇒ Large case–control study using database free of selection bias and generalisable to other high-income settings.
⇒ Exposure well measured from medical systems avoiding recall bias.
⇒ Design accounted for confounding by indication and time-window bias.
⇒ Adjustment for individual socioeconomic status used an ecological proxy and residual confounding possible.
⇒ All secondary brain cancers were combined to maintain statistical power.

development of novel cancer drugs is beset by substantial costs and high failure rates. Repurposing existing approved drugs for different diseases than originally designed potentially saves the considerable time and costs involved in conducting new clinical trials.

Nuclear hormone receptors (NHRs) have crucial roles in cellular homeostasis and have been implicated in the development of cancer. Peroxisome proliferator-activated receptors (PPARs) are a family of ligand-activated transcription factors that regulate the expression of genes and, as a subtype of NHRs, are involved in the control of proliferation and differentiation of cells.[2–4] There are three isoforms of PPARs which are designated as -α, -β/δ and -γ. PPARs have been identified as potential candidates in cancer treatment, with some animal models and early-stage human trials showing beneficial outcomes, including arresting development of precursor cancer cells, decreased serum cancer antigens and tolerance by patients.[5–8] The genes regulated by these transcription

## INTRODUCTION

Individuals diagnosed with primary or secondary (breast, lung, skin melanoma, colorectal and kidney) brain tumours typically have less than a 20% 5-year survival rate and brain tumours are responsible for the most years of life lost of any cancer.[1] The

factors are also involved in the transport, metabolism and storage of fatty acids, in inflammatory processes and in the onset of type 2 diabetes mellitus.[3 9 10]

PPAR-α and -γ are clinically significant due to the use of agonistic compounds such as fibrates in hyperlipidaemia and thiazolidinediones (glitazones) in diabetes, respectively. Both glitazones[11] and fibrates such as fenofibrate cross the blood–brain barrier, although at a slower rate.[12] PPARs are expressed in astrocytes, oligodendrocytes and microglia, and may play a role in growth and differentiation of those cells, including brain tumour stem cells.[13] These drugs, given their safety profile and cost, have the potential for being repurposed for the prevention of brain cancers and may also reduce the risk of secondary tumours by inhibiting angiogenesis.

We hypothesised that PPAR-α and -γ agonists (fibrates and glitazones) would be associated with a reduced risk of developing brain tumours compared with other treatments for hyperlipidaemia and diabetes using the UK Clinical Practice Research Datalink (CPRD), a very large, population-based, primary care database.

## METHODS

We used the CPRD database which contains data from a network of over 2000 general practitioners (GPs) from more than 670 practices across the UK. Validated clinical information is held on over 16 million registered patients, amounting to circa 7% of the UK population[14] and broadly representative of that population.[15 16] Types of data held within the CPRD include, but are not limited to, patient demographics, referrals, hospital admissions, clinical diagnoses and drug prescriptions. Data from secondary care or hospital visits are routinely fed back into the electronic records' system via staff at GP practices. CPRD is jointly sponsored by the Medicines and Healthcare products Regulatory Agency (MHRA) and the National Institute of Health Research (NIHR). Data were extracted from CPRD GOLD and linked to Office for National Statistics (ONS) death registration data and census data on area deprivation (see later).

### Study design and eligible participants

We undertook two nested case–control studies from within the CPRD database as the eligibility criteria (see later) for the fibrate and glitazone analyses were different. All participants in the study had to be aged 18 years or older and registered within the CPRD between 1 January 2000 and 31 December 2016, the former being the first year that glitazones were licensed within the UK. We specified that eligible participants had to be treated with either (i) an anti-hyperlipidaemic or (ii) anti-diabetic medication to reduce the likelihood of "confounding by indication", a type of confounding which can distort the association between drug exposure and outcome when an indication to prescribe a drug (eg, diabetes) is also associated with the outcome. All brain tumour cases and controls that were not treated with either of these classes of drugs

were therefore not included in the sample selection and analyses. The sample size was pragmatic based on all available cases from the dataset rather than based an "a priori" sample size calculation to maximise power.

### Cases

Incident brain tumour (primary or secondary) cases were identified using a large list of Read Codes (online supplemental table 1) and were classified as either of primary or secondary cancers using histopathology.[17] For secondary tumours we did not have information on the site of primary tumour, but we retrospectively searched for another prior cancer diagnosis to estimate the site of the primary cancer. However, if this had been recorded prior to 1 January 2000, or recorded at a non-CPRD participating GP, it may have been missing from our file and was thus not captured by our retrospective search.

### Controls

We sampled controls using "incidence density sampling", whereby on the date each case was diagnosed we identified the potential pool of controls who did not have a prior diagnosis of a brain tumour. Up to four controls were randomly selected per case (to maximise power) and were age-frequency matched to cases by 10-year age groups (<20, 20–29, 30–39, 40–49, 50–59, 60–69, 70–79 and ≥80 years) and male/female sex to ensure reasonable balance of these variables across case–control strata.

Both "immortal time" and "time-window" biases are potential hazards in pharmacoepidemiology studies leading to misleading results,[18] though the former is not relevant to our case–control study design.[19] Time-window bias occurs in case–control studies when the method of selecting controls results in an artefactually longer time window for potential exposure to be identified than their matched cases resulting in a spurious inverse (protective) association between the exposure and outcome.[18] To prevent time-window bias, we specified that controls must have the same retrospective duration of potential exposure (any drug treatment for either hyperlipidaemia or diabetes) within CPRD as the matched case, based on the case index date plus or minus 6 months. For example, if a case had 6.3 years of any drug medication exposure prior to their diagnosis date, then only controls who also had between 5.8 and 6.8 years of any drug exposure over the same secular time period were sampled and hence had the same potential for exposure to either a fibrate or glitazone as cases.

### Exposures

All participants were treated with some medication for either type 2 diabetes or hyperlipidaemia. If they received a fibrate or glitazone they were classified as "exposed" while all the other drugs used for the management of these diseases were classified as the unexposed comparator. Product codes used to determine exposure status are given in online supplemental tables 2 and 3. We excluded participants with type 1 diabetes, defined as those

participants treated solely with insulin therapy, as these patients are very different to those with the more common type 2 diabetes in terms of causes and duration of dysregulated glycaemic control. Participants who started on oral anti-diabetic medication, but at some point in time also received insulin therapy, were eligible for inclusion. Participants on combination therapy were not excluded (eg, both glitazone plus another anti-diabetic medication); these participants were considered as exposed, if one of the therapies was the drug of interest. Similarly, some participants were exposed to both fibrates and glitazones and were considered exposed for both studies; however, we tested both these assumptions in sensitivity analyses (see later).

Exposure was classified in three ways: (i) "Exposed binary" (Yes/No) indicating that a subject has been ever-exposed to a prescription for fibrates or glitazones. (ii) "Longest exposure duration" which was the total uninterrupted time (no breaks of 90 days or more between prescriptions) each participant had been prescribed either fibrates or glitazones. If there was more than one such period, then we took the longest duration. (iii) "Total exposure duration", which was the sum of the total prescription duration (with or without interruptions). Exposure duration for both the longest and total exposure duration were categorised as follows: unexposed, ≤1 year, >1 and ≤2 years, >2 and ≤3 years, >3 and ≤4 years, >4 and ≤5 years, >5 and ≤6 years, >6 and ≤7 years, >7 years. Due to small numbers of participants reaching longer exposure durations, any duration exposure periods over 6 years were considered in the longest duration category in the glitazone analysis, and over 7 years as the longest duration category in the fibrate analysis.

### Potential confounders

The following variables were considered as potential confounders that might influence risk of developing a brain tumour and the choice of drug that a doctor might prescribe: age group, sex and socioeconomic status (SES). We used the Index of Multiple Deprivation 2015 (IMD)[20] as an ecological proxy for individual SES. This is a commonly used measure in the UK that uses census data on a wide variety of economic and health factors to derive a postcode-based deprivation ranked score so that that a higher score indicates less deprivation. IMD scores were grouped into five equal-sized groups from most (quintile 1) to least deprived areas (quintile 5), so we could adjust for any nonlinear associations.

In the glitazone analyses we also tried to adjust for the severity of diabetes as a confounder, as severity may determine choice of anti-diabetic medications and possible cancer risk. We used glycosylated haemoglobin (HbA1c) levels as a measure of glycaemic control and hence severity of diabetes, where available. We converted and standardised the measures to the International Federation of Clinical Chemistry (IFCC) units (mmol/mol). The mean value was calculated using all available HbA1c readings for a subject. A three-level ordinal variable

was created: (1) well-controlled diabetes (≤58 mmol/mol); (2) suboptimally controlled diabetes (>58 mmol/mol and ≤75 mmol/mol); and (3) poorly controlled diabetes (>75 mmol/mol). If there were no measures of the participant's HbA1c levels, we assigned them to the well-controlled group based on the assumption that the clinician did not feel it necessary to monitor the patient's glycaemic control (however, see later for sensitivity analyses that examine this assumption). We attempted to restrict HbA1c readings to only pre-baseline measures; however, due to missing data we had to occasionally include post-baseline HbA1c readings as a proxy measure.

### Statistical methods

Both case–control analyses were conducted using logistic regression to compute odds ratios (ORs) and 95% confidence intervals (CIs) of exposure to fibrate and glitazone drugs versus other drugs used for the same disease. Unadjusted and multivariable models were run, adjusting for the previously mentioned potential confounders, with cases stratified by primary and secondary brain tumours and then combined as all brain tumours if there was no evidence of heterogeneity of effect to enhance statistical power. In addition, we adjusted for total potential exposure duration, defined as the time from either diagnosis or control index date and start of any medication first prescribed for either hyperlipidaemia or diabetes. This was one of the matching variables used to avoid time-window bias. Exposure duration was analysed both as a continuous ordinal variable and also as a "dummy" variable to check for any evidence of nonlinearity in the pattern of the ORs. We used logistic regression rather than conditional logistic models, as the latter may introduce collider bias, and instead we used the matching variables as covariates in the unmatched logistic model as the recommended procedure (see Pearce[21] for detailed justification).

### Sensitivity analyses

The following sensitivity analyses were conducted to check the robustness of our main analyses: (i) Potential selection bias: we reclassified exposure only if the participant's first medication was a fibrate or glitazone. Someone whose first treatment was another drug but was later switched to a glitazone or fibrate in this analysis was classified as unexposed. (ii) Dual exposure: to evaluate if one drug had an effect on risk while the other did not, participants were dropped from the analysis if they were exposed to both a fibrate and a glitazone at the same time. (iii) Latency period: analyses were repeated excluding any exposure for short latencies defined as 6, 12 and 24 months prior to the index date. (iv) Missing HbA1c levels: we used multiple imputation (55 datasets generated, combined using Rubin's rules) to predict missing HbA1c levels based on case–control status, age, IMD, sex, retrospective prescription history, ever-exposure to a glitazone, total glitazone exposure and number of consultations, defined as each day the patient had at least one

in-person consultation. The MICE algorithm in R[22] was used to impute missing HbA1c levels. We also examined whether HbA1c predicted case–control status both in the complete case as well as the imputation dataset. This is because if there was no evidence of an association with HbA1c then it cannot be a confounder and did not need to be included in the multivariable model, avoiding the issue of missing values. Sensitivity analyses were conducted using all brain tumour cases as there was limited statistical power in the separate primary and secondary cancer analyses.

### Patient and public involvement

Patients or the public were not involved in the design, or conduct, or reporting, or dissemination plans of our research.

### RESULTS

For the fibrate analyses there were a total of 1950 cases, of which 1094 were primary and 856 were secondary brain tumours and 7791 matched controls. For the glitazone analyses there were a total of 480 cases, of which 278 were primary and 202 were secondary brain tumours and 1920 controls. There was little evidence of any important differences in age group, sex and IMD distribution between cases versus controls in both the fibrate and glitazone datasets (table 1). Cases were more likely to have well-controlled HbA1c but this was because they were more likely to have missing HbA1c values, which by default were classified as well-controlled diabetes.

There was little evidence that any exposure to fibrates was associated with brain tumour risk for either primary (adjusted OR (aOR) 0.99, 95% CI 0.75 to 1.30; p=0.94), secondary (aOR 0.91, 95% CI 0.67 to 1.23; p=0.55) or all tumours (aOR 0.98, 95% CI 0.78 to 1.24; p=0.88) (table 2). We found little evidence for a dose–response relationship of fibrate exposure, either as longest or total duration, with brain tumour risk for primary, secondary or all tumours (aORs varied from 0.95 to 1.01 and all were consistent with chance). The sensitivity analysis, which only classified exposed if this was the first prescribed anti-hyperlipidaemia medication, found similar results but with less precision (aOR 0.92, 95% CI 0.69 to 1.22; p=0.54)

There was very weak evidence of an inverse association between ever-exposure to glitazones (aOR 0.80, 95% CI 0.65 to 1.07; p=0.14) and all tumours, which was stronger with secondary tumours (aOR 0.68, 95% CI: 0.46, 1.02; p=0.06) rather than primary tumours (aOR 0.96, 95% CI 0.69 to 1.33; p=0.81) (table 3), though this could have been due to chance. The inverse association was stronger and consistent across both primary and secondary tumours when the exposure was classified as longest duration; primary (aOR per year 0.89, 95% CI 0.80 to 0.98; p=0.02), secondary (aOR per year 0.87, 95% CI 0.77 to 0.99; p=0.02) and all brain tumours (aOR per year 0.88, 95% CI 0.81 to 0.95; p=0.002). Very similar results were

seen with total duration of exposure (eg, aOR per year for all tumours 0.85, 95% CI 0.79 to 0.92; p=$8.00\times10^{-5}$).

### Sensitivity analyses

Results from the analysis of duration period as a categorical, instead of a continuous, variable found that the inverse association with glitazone and all tumours only emerged after 3 years of exposure; however, the confidence intervals for the shorter duration periods were sufficiently wide as to not exclude any dose–response effect across time (see online supplemental table 4). When we only classified exposed if this was the first prescribed anti-diabetes medication we found little evidence of an association but with very wide confidence intervals due to fewer exposed cases and controls (aOR 1.10, 95% CI 0.35 to 3.44; p=0.87) (see online supplemental table 4). Effects were similar (longest duration aOR 0.85, 95% CI 0.78 to 0.93; p=$4.00\times10^{-4}$) when investigating the effects of monotherapy.

### Investigating missing HbA1c levels

HbA1c results were missing for 54.8% of subjects in the glitazone analysis. We observed similar results for the association of both glitazone exposure status (aOR 0.81, 95% CI 0.66 to 1.00; p=0.11) and longest duration of glitazone exposure (aOR per year 0.87, 95% CI 0.82 to 0.93; p=0.001) with all brain tumour risk when multiple imputation was used for the missing HbA1c values (online supplemental table 5). There was little evidence of an association between poorly versus well-controlled HbA1c and case–control status in both the imputed (aOR 0.95, 95% CI 0.69 to 1.32; p=0.80) and complete case analysis (aOR 1.02, 95% CI 0.60 to 1.73; p=0.80), suggesting that HbA1C may not be a confounder as it did not predict case–control status and therefore adjustment may not be necessary. This observation was supported by the aOR without adjustment for HbA1c, which was almost identical to the imputed results so that the OR per year of glitazaone exposure with longest duration was 0.87 (95% CI 0.80 to 0.94; p=0.0008) (online supplemental table 5).

### Latency period

Similar associations of both fibrates and glitazones with all brain tumour risk were observed in the sensitivity analyses with different latency periods. In the glitazone analysis the inverse association seen in the initial analyses for longest duration was very similar as the latency period increased (aORs after a 6-, 12- and 24-month latency were 0.88, 0.87 and 0.83, respectively) and unlikely to be due to chance (online supplemental table 6).

### DISCUSSION

To the best of our knowledge, this is the first pharmaco-epidemiological study of the effects of fibrate and glitazone prescription on risk of brain tumours compared with other treatments for hyperlipidaemia and type 2 diabetes. There have been various in vivo studies that

**Table 1**  Distribution of possible confounders by case–control status for both fibrate and glitazone datasets

| Variable | Cases (n (%)) | Controls (n (%)) | Crude OR (95% CI) | P value |
|---|---|---|---|---|
| **Fibrates case–control** | **(N=1950)** | **(N=7791)** | | |
| Male | 1118 (57.3) | 4469 (57.4) | Referent | – |
| Female | 832 (42.7) | 3322 (42.6) | 1.00 (0.91 to 1.11) | 0.98 |
| Age, years | | | | |
| <20 | 0 (0.0) | 0 (0.0) | – | – |
| 20–29 | 0 (0.0) | 0 (0.0) | – | – |
| 30–39 | 2 (0.1) | 5 (0.1) | 1.58 (0.31 to 8.17) | 0.58 |
| 40–49 | 6 (0.3) | 29 (0.4) | 0.82 (0.34 to 1.98) | 0.66 |
| 50–59 | 40 (2.1) | 193 (2.5) | 0.82 (0.58 to 1.16) | 0.26 |
| 60–69 | 220 (11.3) | 848 (10.9) | 1.03 (0.87 to 1.21) | 0.75 |
| 70–79 | 561 (28.8) | 2279 (29.3) | 0.97 (0.87 to 1.09) | 0.65 |
| ≥80 | 1121 (57.5%) | 4437 (57.0%) | Referent | – |
| IMD, quintile | | | | |
| 1, lower SES | 457 (23.4) | 1768 (22.7) | 1.05 (0.91 to 1.22) | 0.48 |
| 2 | 439 (22.5) | 1790 (23.0) | Referent | – |
| 3 | 430 (22.1) | 1594 (20.5) | 1.10 (0.95 to 1.28) | 0.21 |
| 4 | 331 (17.0) | 1466 (18.8) | 0.92 (0.79 to 1.08) | 0.31 |
| 5, higher SES | 293 (15.0) | 1173 (15.1) | 1.02 (0.86 to 1.20) | 0.83 |
| **Glitazones case–control** | **(N=480)** | **(N=1920)** | | |
| Male | 277 (57.7) | 1108 (57.7) | Referent | – |
| Female | 203 (42.3) | 812 (42.3) | 1.00 (0.82 to 1.22) | 1.00 |
| Age, years | | | | |
| <20 | 1 (0.2) | 1 (0.1) | 3.93 (0.24 to 63.00) | 0.33 |
| 20–29 | 1 (0.2) | 7 (0.4) | 0.56 (0.07 to 4.58) | 0.59 |
| 30–39 | 3 (0.6) | 11 (0.6) | 1.07 (0.30 to 3.87) | 0.92 |
| 40–49 | 7 (1.5) | 28 (1.5) | 0.98 (0.42 to 2.27) | 0.97 |
| 50–59 | 15 (3.1) | 76 (4.0) | 0.78 (0.44 to 1.37) | 0.38 |
| 60–69 | 56 (11.7) | 233 (12.1) | 0.94 (0.68 to 1.30) | 0.73 |
| 70–79 | 135 (28.1) | 535 (27.9) | 0.99 (0.79 to 1.25) | 0.94 |
| ≥80 | 262 (54.6) | 1029 (52.6) | Referent | – |
| IMD, quintile | | | | |
| 1, lower SES | 90 (18.8) | 362 (18.9) | 1.33 (0.95 to 1.86) | 0.09 |
| 2 | 114 (23.8) | 376 (19.6) | 1.62 (1.18 to 2.24) | 0.003 |
| 3 | 109 (22.7) | 384 (20.0) | 1.52 (1.10 to 2.10) | 0.01 |
| 4 | 78 (16.3) | 418 (21.8) | Referent | – |
| 5, higher SES | 89 (18.5) | 380 (19.8) | 1.26 (0.90 to 1.75) | 0.18 |
| Mean prescription history, years (SD) | 2.12 (2.41) | 1.99 (2.27) | 1.02 (0.98 to 1.07)* | 0.27 |
| HbA1c* | | | | |
| 1, well-controlled | 418 (87.0) | 1490 (77.6) | Referent | – |
| 2 | 42 (9.0) | 305 (15.9) | 0.49 (0.35 to 0.69) | 0.001 |
| 3, poorly controlled | 20 (4.0) | 125 (6.5) | 0.57 (0.35 to 0.93) | 0.02 |
| Missing (coded as 1) | 338 (70.4) | 979 (51.0) | – | – |

*HbA1C, glycosylated haemoglobin, a measure of glycaemic control used to monitor patients with diabetes.
CI, confidence interval; HbA1c, glycosylated haemoglobin; IMD, Index of Multiple Deprivation 2015; OR, odds ratio; SD, standard deviation; SES, socioeconomic status.

**Table 2** Case–control results for primary, secondary and all brain tumour cases with fibrate exposure

| Variable | Primary brain tumours | | | | Secondary brain tumours | | | | All brain tumours | | | |
|---|---|---|---|---|---|---|---|---|---|---|---|---|
| | Cases (n (%)) | Controls (n (%)) | aOR (95% CI)* | P value | Cases (n (%)) | Controls (n (%)) | aOR (95% CI)* | P value | Cases (n (%)) | Controls (n (%)) | aOR (95% CI)* | P value |
| | (N=1094) | (N=4375) | | | (N=856) | (N=3416) | | | (N=1950) | (N=7791) | | |
| Fibrate exposure status | | | | | | | | | | | | |
| Exposed | 68 (6.2) | 278 (6.4) | 0.99 (0.75 to 1.30) | 0.94 | 56 (6.5) | 245 (7.2) | 0.91 (0.67 to 1.23) | 0.55 | 124 (6.4) | 523 (6.7) | 0.98 (0.78 to 1.24) | 0.88 |
| Unexposed | 1026 (93.8) | 4097 (93.7) | Referent | – | 800 (93.5) | 3171 (92.8) | Referent | – | 1826 (93.6) | 7268 (93.3) | Referent | – |
| Fibrate longest duration, days (SD)† | 799 (1050) | 755 (913) | 1.01 (0.94 to 1.09) | 0.76 | 676 (913) | 878 (1123) | 0.95 (0.87 to 1.03) | 0.16 | 743 (989) | 813 (1018) | 0.97 (0.91 to 1.03) | 0.35 |
| Fibrate total duration, days (SD)† | 969 (1183) | 1073 (1233) | 0.99 (0.93 to 1.05) | 0.64 | 871 (1232) | 1217 (1408) | 0.95 (0.88 to 1.01) | 0.08 | 925 (1202) | 1141 (1319) | 0.97 (0.92 to 1.01) | 0.11 |

*Model adjusted for sex, age, Index of Multiple Deprivation 2015, retrospective prescription history (categorised yearly).
†Odds ratio calculated for per year increase in variable.
aOR, adjusted odds ratio; CI, confidence interval; SD, standard deviation.

have suggested fibrate exposure may be protective for brain tumours, specifically gliomas, by modulating PPAR-α inhibition.[23–26] There is similar evidence available for the consideration of glitazones as a treatment option from in vivo studies.[27–30] However, prior to this study there have been no other pharmacoepidemiological studies that examined whether fibrates or glitazones affect brain tumour risk.

In the glitazone case–control analysis there was a modest inverse effect observed for ever-exposure to glitazones, which was seen more strongly in the dose–response analyses, especially after exposure durations of four or more years, in both univariable and multivariable models. Repeating our analyses and accounting for various latency periods did not alter our conclusions. Similar inverse associations were seen for both primary and secondary brain tumours, but limited data meant analyses could not test whether there were differences by type of primary cancer for those subjects with secondary brain tumours.

We found little evidence that exposure to fibrates was associated with all, primary or secondary brain tumour risk, in contrast to some previous literature.[31] Sensitivity analyses, investigating a potential dose–response effect, or exposure status based on whether fibrates were the first prescribed drug for treatment, also showed little evidence of an association between fibrate exposure and all brain tumour risk.

The assumption that missing HbA1c levels should be assumed to reflect good glycaemic control was tested using follow-up sensitivity analyses to ensure this did not bias the potential negative association between glitazone exposure and brain tumour risk. However, in both the multiple imputation and complete case analysis we found no association between HbA1c and case–control status. Furthermore, the results using both multiple imputation or no adjustment were consistent with the primary analyses.

This study has several strengths and limitations. We used a nested case–control design within CPRD, a large and well-established database which has been validated by numerous sources.[15 16 32] It should be free of selection bias as almost all UK residents are registered with a GP and the population captured by CPRD-registered practices is representative of the general population.[14] Exposure was well measured as it was recorded from medical systems that used the record to print out prescribed drugs for redemption at a pharmacist. The exposure data were collected prospectively, prior to diagnosis, which avoids recall bias. However, we have only measured receipt of a prescription and have no data on actual drug intake or therapeutic levels. Despite this, as both the response to therapy are monitored, the dosage is likely to be adjusted in response to good or bad control of the subjects' lipids and blood sugar levels. Exposure was classified from first prescription of any drug for lipids or diabetes and controls were sampled to have similar total prescription duration to avoid time-window bias, which was also adjusted in the multivariable models. The risk of confounding by

**Table 3** Case–control results for primary, secondary and all brain tumour cases with glitazone exposure

| Variable | Primary brain tumours | | | | Secondary brain tumours | | | | All brain tumours | | | |
|---|---|---|---|---|---|---|---|---|---|---|---|---|
| | Cases (n (%)) | Controls (n (%)) | aOR (95% CI)* | P value | Cases (n (%)) | Controls (n (%)) | aOR (95% CI)* | P value | Cases (n (%)) | Controls (n (%)) | aOR (95% CI)* | P value |
| | (N=278) | (N=1112) | | | (N=202) | (N=808) | | | (N=480) | (N=1920) | | |
| Glitazone exposure status | | | | | | | | | | | | |
| Exposed | 61 (21.9) | 261 (23.5) | 0.96 (0.69 to 1.33) | 0.81 | 36 (17.8) | 199 (24.6) | 0.68 (0.46 to 1.02) | 0.06 | 97 (20.2) | 460 (24.0) | 0.80 (0.65 to 1.07) | 0.14 |
| Unexposed | 217 (78.1) | 851 (76.5) | Referent | – | 166 (82.2) | 609 (75.4) | Referent | – | 383 (79.8) | 1460 (76.0) | Referent | – |
| Glitazone longest duration, days (SD)† | 551 (599) | 851 (754) | 0.89 (0.80 to 0.98) | 0.02 | 701 (550) | 899 (767) | 0.87 (0.77 to 0.99) | 0.02 | 607 (583) | 872 (760) | 0.88 (0.81 to 0.95) | 0.002 |
| Glitazone total duration, days (SD)† | 684 (695) | 1000 (809) | 0.90 (0.82 to 0.99) | 0.02 | 845 (681) | 1073 (842) | 0.88 (0.79 to 0.98) | 0.01 | 603 (720) | 1002 (823) | 0.86 (0.79 to 0.92) | 0.00008 |

*Model adjusted for sex, age, Index of Multiple Deprivation 2015, retrospective prescription history (categorised yearly) and mean glycosylated haemoglobin (HbA1c) levels.
†Odds ratio calculated for per year increase in variable.
aOR, adjusted odds ratio; CI, confidence interval; SD, standard deviation.

indication was minimised by only sampling cases and controls who could have potentially been exposed to fibrates or glitazones because of a clinical indication. However, the consequence of this sampling strategy is that the number of available cases was markedly reduced with reduced statistical power to detect modest effects. Survivor bias is unlikely as all cases were included from diagnosis regardless of subsequent life expectancy and there is no evidence that the exposures themselves effect mortality risk. Not all potential risk factors for brain tumours were captured by CPRD (eg, previous irradiation) but if these are not associated with exposure this will not confound the results. Adjustment for SES was at an ecological rather than individual level but if anything is likely to bias the associations to the null due to non-differential measurement error. We had to combine all secondary brain cancers regardless of the primary cancer due to limited sample size so that there would have been no power to test the hypothesis for specific types of secondary brain tumours. Many of the secondary brain tumour cases had a previous diagnosis of common cancers which metastasise to brain, in particular breast, lung and skin cancers.

Overall longer glitazone exposure was associated with a reduced risk of brain metastases from these primary sites but larger studies should be conducted to determine whether the association of glitazone exposure and secondary brain tumours is similar or not across these different primary sites. Risk of brain tumours may be greater in populations with either type 2 diabetes or hyperinsulinaemia[33 34] though further studies are required to confirm these associations. It is therefore possible that glitazone may reduce risk of both primary and/or secondary brain tumours by improved glycaemic control. In addition, mice injected with Lewis lung cancer which had been resected were less likely to develop metastatic disease after being treated with rosiglitazone compared with placebo, possibly by its anti-angiogenic effect.[35] We cannot say if we would have observed the same associations if glitazones were prescribed as a preventative therapy in a high-risk non-diabetic population as the generalisability of our results are limited to those brain tumour cases with type 2 diabetes.

## CONCLUSIONS
Longer exposure to glitazones was inversely associated with diagnosis of primary and secondary brain tumours, but we found little evidence of an association between fibrates and any type of brain tumour. Our results should be generalisable to other high-income settings but may not be generalisable to low–middle-income populations. Further research is required to investigate whether these findings are replicated using independent datasets, preferably larger in size and/or with better data on glycaemic control and other potential confounders. If the glitazone association is biologically causal, this may lead to a better understanding of pathophysiological mechanisms and potential therapies for the prevention of brain cancers.

For example, patients with specific aggressive cancers that have a high probability of brain metastases could be treated with glitazones as a tertiary prevention strategy to prevent or delay secondary spread to the brain. Such a hypothesis could be tested in a future double-blind clinical trial if stronger evidence emerges from other studies, given the safety and current use of glitazones for the management of diabetes.

### Ethics approval
The study was approved by the Independent Scientific Advisory Committee (ISAC) for the UK Medicines and Healthcare Products Regulatory Agency (ISAC Protocol Number: 18_149R). Participant consent is not required for ISAC approved Clinical Practice Research Datalink (CPRD) studies using purely observational data. We assert that all procedures contributing to this work comply with the ethical standards of the relevant national and institutional committees on human experimentation and with the Helsinki Declaration of 1975, as revised in 2008.

**Author affiliations**
[1]MRC Integrative Epidemiology Unit, University of Bristol, Bristol, UK
[2]Department of Population Health Sciences, University of Bristol Medical School, Bristol, UK
[3]National Institute for Health Research (NIHR) Bristol Biomedical Research Centre, University Hospitals Bristol and Weston NHS Foundation Trust and University of Bristol, Bristol, UK
[4]Population, Policy and Practice Research and Teaching Department, UCL Great Ormond Street Institute of Child Health, London, UK
[5]National Institute for Health Research (NIHR) Collaboration for Leadership in Applied Health Research and Care (CLAHRC) West, Univeristy of Bristol, Bristol, UK
[6]Brain Tumour Research Centre, University of Bristol, Bristol, UK

**Contributors** Study hypothesis and design: KMK, RM, YB-S. Extraction of Clinical Practice Research Datalink (CPRD) data: MMCE, MTR. Conceptualisation of analysis plan: RM, YB-S, KMK, MO, MTR. Performed data management and statistical analyses: JWR, MO. Writing of the initial versions of the manuscript: JWR followed by YB-S. Responsible for the overall content as the guarantor: YB-S. All authors read, reviewed and provided feedback for the final manuscript.

**Funding** The Medical Research Council (MRC)/University of Bristol Integrative Epidemiology Unit (IEU) is supported by the MRC and the University of Bristol (MC_UU_12013/1, MC_UU_12013/2). JWR is supported by a joint studentship from NHS Southmead Hospital Charity (SOCS/SJ1447). RMM is supported by a Cancer Research UK (CRUK) Programme Grant, the Integrative Cancer Epidemiology Programme (C18281/A29019). JWR and RMM are members of the MRC IEU which is supported by the MRC and the University of Bristol (MC_UU_12013/1-9). RMM is supported by the National Institute for Health Research (NIHR) Bristol Biomedical Research Centre which is funded by the NIHR and is a partnership between University Hospitals Bristol NHS Trust and Weston NHS Foundation Trust and the University of Bristol. RMM is a NIHR Senior Investigator (NIHR202411). This work was supported by funding from the Brain Tumour Bank and Research Fund, Southmead Hospital Charity (Registered Charity Number: 1055900), and University of Bristol Campaigns and Alumni funding. This work was supported by CRUK (Grant Number: C18281/A29019), as part of the Integrative Cancer Epidemiology Programme. MTR, MMCE and YBS are supported by the NIHR Applied Research Collaboration West (NIHR ARC West) (YB-S also receives HEFCE funding). The views expressed in this article are those of the author(s) and not necessarily those of the NIHR or the Department of Health and Social Care.

**Competing interests** JWR receives funding from Biogen for unrelated research. All other authors declare no conflicts of interest.

**Patient and public involvement** Patients and/or the public were not involved in the design, or conduct, or reporting, or dissemination plans of this research.

**Patient consent for publication** Not applicable.

**Ethics approval** This study involves human participants and was approved by the Independent Scientific Advisory Committee (ISAC) for the UK Medicines and Healthcare Products Regulatory Agency (ISAC Protocol Number: 18_149R). Participant consent is not required for ISAC-approved Clinical Practice Research Datalink (CPRD) studies using purely observational data.

**Provenance and peer review** Not commissioned; externally peer reviewed.

**Data availability statement** Data may be obtained from a third party and are not publicly available. The Clinical Practice Research Datalink (CPRD) is an electronic healthcare record database open to all researchers. Researchers can apply to access CPRD data and, if successful, can access the data of their choosing. The CPRD charges researchers and other organisations to access this data. The data that support the findings of this study are available from the CPRD but restrictions apply to the availability of these data, which were used under license for the current study, and so are not publicly available.

**ORCID iDs**
Martha Maria Christine Elwenspoek http://orcid.org/0000-0002-9824-9335
Maria Theresa Redaniel http://orcid.org/0000-0002-0668-0874
Yoav Ben-Shlomo http://orcid.org/0000-0001-6648-3007

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
