## [Reviewer comments · BMJ Open]

ARTICLE DETAILS

TITLE (PROVISIONAL)	Use of drugs for hyperlipidaemia and diabetes and risk of primary and secondary brain tumours: nested case control studies using the UK Clinical Practice Research Datalink (CPRD).
AUTHORS	Robinson, Jamie W; Martin, Richard; Ozawa, Mio; Elwenspoek, Martha; Redaniel, Maria Theresa; Kurian, Kathreena; Ben-Shlomo, Yoav

VERSION 1 – REVIEW

REVIEWER	Bunevicius, Adomas Hospital of Lithuanian University of Health Sciences Kauno klinikos, Neurosurgery
REVIEW RETURNED	13-Mar-2023

GENERAL COMMENTS	• The concept of primary brain tumors is not clearly emphasized in the paper as they can range from malignant gliomas with poor survival to low grade gliomas and meningiomas that typically have more favorable prognosis. Consideration of various primary brain tumors into a single category is not entirely clear given heterogeneity of the disease.• The rationale for consideration of secondary or metastatic brain tumors is not clearly justified in the paper because brain metastases represent a progression of primary cancer rather than intrinsic brain disease. Therefore, the rationale behind considering both primary and secondary brain tumors is not entirely clear. It is not clear why the rated mechanisms should be implicated in the incidence of different diseases.• When considering metastatic brain tumors it is important to consider cancer type given differences in epidemiology of metastatic brain tumors.• Was medication of interest exposure timing with regards to brain tumor diagnosis considered in the analyses?• Consider reporting distribution of types of different brain tumors in the study cohort. Authors should consider sensitivity analyses by brain tumor types to better understand the observed associations.• Authors discuss studies regarding the potential use of medication of interest for cancer treatment, however it appears that the study was more focused on the preventive role of these treatments.• Unclear why the results cannot be generalizable to low income settings (Conclusions section).
--

REVIEWER	Scicchitano, Pietro ASL Bari, Cardiology
REVIEW RETURNED	15-Mar-2023

GENERAL COMMENTS	Dear Editor,
--------------

	I read this paper and I think that:  - The retrospective nature of this study might be considered as a limitation. Please discuss such a point in a dedicated limitation section. Please provide. - Indeed, cancer occurrence is the mix of several factors. For example, environmental conditions and exposure to environmental contaminants might impact on results. These data are missing. Authors should have included these data and computed them in the final analysis. - Similar consideration is the need for identifying occupational activities of patients. These conditions might condition the occurrence of brain cancers. Please discuss such a point. - A worrisome point is the inclusion of “secondary brain tumors”:  a- First, authors should indicate the origin and the type of the “primary mass”: different types and site of origin of masses might differently impact on the occurrence of brain metastases independently from other drug therapies. b- Secondly, authors should specify type of treatments of the original cancer: chemo or radiotherapy or both with their related dosages and/or site of irradiation. This is important for confirming the final results of the paper or not. - Fibrates and antilipemic drugs – as well as the entire world of drugs – might provoke adverse effects thus leading to drug discontinuation, changing in type of molecules, reduction in dosages. All of these conditions might impact on final results. Data should be provided. - It is not clear the follow-up period. Authors should specifically address the follow-up period issue as longer follow-ups do implicate too many confounding factors which might negatively impact on final results.
--	---

REVIEWER	Steinke, Douglas The University of Manchester, Centre for Pharmacoepidemiology and Drug Safety
REVIEW RETURNED	12-Jun-2023

GENERAL COMMENTS	This is a well written paper and presents an important issue in repurposing medicines for use in other indications. The authors present a theoretical association between fibrates and glitazones and brain tumours. However using RWD, they do not find the association. Only long term use shows some relationship, but this could be survivor bias. Confounding and bias was accounted for as mentioned in the study. This a well designed study with a sufficient sample size to obtain a definitive result.
--

REVIEWER	Suzuki, Shinya National Cancer Center-Hospital East, Department of Pharmacy
REVIEW RETURNED	04-Nov-2023

GENERAL COMMENTS	This manuscript is an important update report of a well-designed case-control study of glitazone or fibrate use and risk of primary and secondary brain tumors using the CPRD to clarify the hypothesis from basic research that long-term exposure to glitazones is associated with a decreased risk of primary and secondary brain tumors. I believe the description of the methods and results and the discussion to the conclusion are well described and worthy of inclusion in BMJ-Open. One minor point, please correct the lack of explanation for the ** on HbA1c** in Table 1 and the absence of *.
---

VERSION 1 – AUTHOR RESPONSE

Reviewer: 1

Dr. Adomas Bunevicius, Hospital of Lithuanian University of Health Sciences Kauno klinikos
Comments to the Author:

- The concept of primary brain tumors is not clearly emphasized in the paper as they can range from malignant gliomas with poor survival to low grade gliomas and meningiomas that typically have more favorable prognosis. Consideration of various primary brain tumors into a single category is not entirely clear given heterogeneity of the disease.

We fully agree with this statement and this would apply to any other sort of cancer. However, we had no prior evidence to suggest that the therapeutic benefits would be seen with any specific type of primary tumour and hence it made sense to treat them as a single group. If we had looked at specific sub-groups, not only would we have had less power, but we would have been accused by reviewers of data dredging to find a “type 1” chance statistically significant finding. The reviewer fails to consider that by combining all tumours we have, if anything diluted our chance of observing a benefit assuming there is only benefit in some sub-types. Hence our observation of glitazones being associated with a reduced risk is on the conservative side and warrants further investigation.

- The rationale for consideration of secondary or metastatic brain tumors is not clearly justified in the paper because brain metastases represent a progression of primary cancer rather than intrinsic brain disease. Therefore, the rationale behind considering both primary and secondary brain tumors is not entirely clear. It is not clear why the rated mechanisms should be implicated in the incidence of different diseases.

This is a very reasonable point which is why we examined the two types of tumours separately to see if there was any evidence of heterogeneity. We agree that the prior basic science evidence is more focussed on primary brain tumours however it is biologically plausible that these agents could reduce secondary brain metastases through the inhibition of angiogenesis. For example, Paniograhay et al JCI doi:10.1172/JCI200215634 showed, using the Lewis lung cancer model in mice, that treatment with rosiglitazone compared to control prevented invasive metastases. Furthermore better treatment of insulin resistance in diabetics may be associated with reduced risk of other primary cancers (excluding brain tumours) which in turn would reduce risk of secondary brain tumours. We would be happy to expand on this rationale if we are invited to resubmit the paper.

- When considering metastatic brain tumors it is important to consider cancer type given differences in epidemiology of metastatic brain tumors.

We fully agree and we tried to do this in our paper. Most of our secondary cancers originated from breast, lung and skin but our sample size limited us from exploring these sub-groups further as we were under-powered.

- Was medication of interest exposure timing with regards to brain tumor diagnosis considered in the analyses?

Yes- All exposure timing and duration were measured prior to diagnosis of cases and controls using incidence density sampling and selecting the same exposure time window to avoid this bias (explained in the methods section). As can be seen in supplemental table 4 we present the odds ratios for increasing duration of glitazone exposure.

- Consider reporting distribution of types of different brain tumors in the study cohort. Authors should consider sensitivity analyses by brain tumor types to better understand the observed associations.

We present all the main analyses stratified by primary and secondary tumours. The staff member who led on the analysis (JWR) has now left the University making it harder to run

analyses as they wrote all the code but we could try to report the frequency of brain tumour sub-groups if this is felt to be important by the editorial team. Please note that clinically there is often no definitive diagnosis as often the tumour is diagnosed on neuroimaging and it is felt not appropriate to get tissue diagnosis confirmation.

• Authors discuss studies regarding the potential use of medication of interest for cancer treatment, however it appears that the study was more focused on the preventive role of these treatments.

We think the reviewer has been confused by the use of the term tertiary prevention and we respectfully disagree with his comment. We stated in the discussion “patients with specific aggressive cancers that have a high probability of brain metastases, could be treated with glitazones as a tertiary prevention strategy to prevent or delay secondary spread to the brain”. In Public Health jargon “tertiary prevention” refers to treating individuals with diagnosed disease, in this case cancer, to reduce mortality/morbidity. This is not the same as treating people without disease (primary prevention) or with risk factors for diseases (e.g. treating HBP to reduce risk of future cardiovascular disease) (secondary prevention).

• Unclear why the results cannot be generalizable to low income settings (Conclusions section).

It is fairly standard to be cautious in extrapolating results from high to low income settings. For example, there may or may not be genetic or other mechanistic differences in pathophysiology. For example type II diabetes is far more common in the South Asian continent and it is possible that other factors influence or interact with the biological effects of glitazones. On a simple pragmatic level there may access and economic factors in some LMIC settings.

Reviewer: 2

Dr. Pietro Scicchitano, ASL Bari

Comments to the Author:

I read this paper and I think that:

- The retrospective nature of this study might be considered as a limitation. Please discuss such a point in a dedicated limitation section. Please provide.

This comment displays some misunderstanding around the study design. All case control studies are by design “retrospective” whilst cohort studies may be either prospective or retrospective. The reviewer is correct to be cautious when the exposure is measured by self-report in a case control study with the inherent danger of “recall bias”. However, this is not an issue as our study is nested within the CPRD cohort and our exposure, specific medications, are very well recorded for medico-legal reasons as they are doctor prescribed drugs. Hence this comment is simply not valid.

- Indeed, cancer occurrence is the mix of several factors. For example, environmental conditions and exposure to environmental contaminants might impact on results. These data are missing. Authors should have included these data and computed them in the final analysis.

This statement is probably true of the vast majority of diseases. One does not need data on all the known causal determinants of a disease to identify a new cause. What is of more concern is the issue of confounding which we discuss in the paper. If editors followed the advice of this reviewer that “Authors should have included these data and computed them in the final analysis” no observational study would ever be published yet these are essential to provide the evidence base for follow-on RCTs, where appropriate.

- Similar consideration is the need for identifying occupational activities of patients. These conditions might condition the occurrence of brain cancers. Please discuss such a point.

To the best of our knowledge there are no known occupational causes of primary brain tumours but even if there were this would only be of relevance if this itself predicted use of glitazones as it would be a confounder. See comments above.

- A worrisome point is the inclusion of “secondary brain tumors”:

We do not agree. We feel it is actually helpful to examine if the associations are seen, or not, for both primary and secondary brain tumours (see comments above). We simply do not understand how it is a disadvantage to present evidence on both primary and secondary brain tumours.

b- First, authors should indicate the origin and the type of the “primary mass”: different types and site of origin of masses might differently impact on the occurrence of brain metastases independently from other drug therapies.

We report this in the discussion. They are mainly breast, lung and skin but we do not have ideal data on this. See comments above in relation to why we cannot undertake further sub-group analyses

b- Secondly, authors should specify type of treatments of the original cancer: chemo or radiotherapy or both with their related dosages and/or site of irradiation. This is important for confirming the final results of the paper or not.

This comment suggest that reviewer misunderstand the nature of our analysis which is to test an aetiological hypothesis. We are looking at exposures prior to diagnosis in cases and controls. Hence the treatment of the cancer (which only applies to cases) has no role in this analysis and can in no way confirm or refute the results.

- Fibrates and antilipemic drugs – as well as the entire world of drugs – might provoke adverse effects thus leading to drug discontinuation, changing in type of molecules, reduction in dosages. All of these conditions might impact on final results. Data should be provided.

Of course all drugs can be discontinued for side effects. Whilst we do not specifically report discontinuation data, this aspect is included in our detailed description of exposure classification into “longest and total exposure duration”. Hence if someone stops a drug and then restarts it or another drug in the same class this is included in total exposure and has therefore been accounted for by comparing these two exposure classifications.

- It is not clear the follow-up period. Authors should specifically address the follow-up period issue as longer follow-ups do implicate too many confounding factors which might negatively impact on final results.

Again the reviewer has failed to understand that this is a nested case control study and not a cohort study. Hence there is no follow-up period as we start with cases and controls and go back in time. Furthermore they simply misunderstand the issue of long follow-up. This relates to attrition (loss to follow-up bias) and not to confounding per se. What is true is the status of a confounder may change over time (this is known as a time-varying confounder) but I suspect this is not what they meant.

Reviewer: 3

Dr. Douglas Steinke, The University of Manchester

Comments to the Author:

This is a well written paper and presents an important issue in repurposing medicines for use in other indications. The authors present a theoretical association between fibrates and glitazones and brain tumours. However using RWD, they do not find the association. Only long term use shows some relationship, but this could be survivor bias. Confounding and bias was accounted for as mentioned in the study. This a well designed study with a sufficient sample size to obtain a definitive result.

We thank the reviewer for his positive comments. We added a comment about survivor bias in the discussion though think it is an unlikely explanation for our observation.

Reviewer: 4

Dr. Shinya Suzuki, National Cancer Center-Hospital East

Comments to the Author:

This manuscript is an important update report of a well-designed case-control study of glitazone or fibrate use and risk of primary and secondary brain tumors using the CPRD to clarify the hypothesis from basic research that long-term exposure to glitazones is associated with a decreased risk of primary and secondary brain tumors.

I believe the description of the methods and results and the discussion to the conclusion are well described and worthy of inclusion in BMJ-Open. One minor point, please correct the lack of explanation for the ** on HbA1c** in Table 1 and the absence of *.

We thank the reviewer for their positive comments and have corrected the * as well as adding an explanation of HbA1C in table 1